# Young Smokers’ Therapy Preferences: App-Based vs. Face-to-Face Treatment in the Context of Co-Addictions

**DOI:** 10.3390/healthcare13182326

**Published:** 2025-09-17

**Authors:** Francisca López-Torrecillas, María del Mar Arcos-Rueda, Beatriz Cobo-Rodríguez, Lucas Muñoz-López

**Affiliations:** 1Departamento de Personalidad, Evaluación y Tratamiento Psicológico, Universidad de Granada, 18011 Granada, Spain; fcalopez@ugr.es; 2Facultad de Ciencias, Hospital Universitario de la Paz, 18071 Granada, Spain; mar.ar@live.com; 3Departamento de Métodos Cuantitativos para la Economía y la Empresa, Universidad de Granada, 18011 Granada, Spain; beacr@ugr.es

**Keywords:** smoking cessation, app-based therapy, face-to-face therapy, young adults, decisional balance, comorbid addictions

## Abstract

**Background:** Tobacco use remains a major public health concern among young adults and is often complicated by co-occurring addictive behaviors. **Objective:** This study analyzed motivation for change, assessed with the decisional balance framework, in relation to multiple addictions among young smokers seeking treatment. **Methods:** Ninety-eight participants from the University of Granada enrolled in either an app-based cognitive–behavioral therapy (CBT) program (n = 35) or a traditional face-to-face CBT program (n = 63). Recruitment relied on self-identification and voluntary participation. Standardized instruments were applied to measure nicotine dependence (FTND), behavioral and substance-related addictions (MULTICAGE CAD-4), cannabis dependence (SDS), and motivation for change (DBQ). Logistic and stepwise regression analyses were conducted to identify predictors of treatment choice and motivational outcomes. **Results:** Younger participants and students were more likely to choose the app-based program. Compulsive buying was linked to perceiving more disadvantages of smoking, whereas sex addiction, cannabis dependence, and other substance addictions were associated with perceiving fewer disadvantages. **Conclusions**: Treatment preferences and motivational profiles differ according to age, academic status, and co-occurring addictions. These findings highlight the need to tailor smoking cessation strategies to individual profiles and support the role of mobile health tools in engaging digitally oriented populations.

## 1. Introduction

Tobacco consumption remains a global public health concern and continues to be one of the leading preventable causes of morbidity and mortality [1]. Despite declining smoking rates in some high-income countries, tobacco use persists at alarming levels, particularly among adolescents and young adults [2]. The risks include cardiovascular disease, chronic respiratory conditions, and cancer, which create not only individual harm but also a major societal and economic burden [1,2].

Traditional smoking cessation programs for young people often involve individual or group behavioral interventions, cognitive–behavioral therapy (CBT), school-based prevention initiatives, and telephone quitlines [3,4,5]. Over the past decade, digital health tools such as mobile applications and web-based platforms have been increasingly incorporated into cessation strategies [2,6]. These tools offer flexibility, accessibility, and continuous support, making them especially appealing to younger smokers familiar with digital environments [7,8]. However, adherence remains a major challenge, as many young individuals attempt to quit without professional help or discontinue programs prematurely [9,10,11].

Understanding motivational factors is, therefore, critical for improving engagement in cessation programs. Prior studies indicate that age, education, nicotine dependence, and previous quit attempts predict treatment initiation and adherence [12,13,14]. Psychological aspects are equally important, particularly motivation for change. This concept, central to the transtheoretical model of change, is often assessed through decisional balance, which reflects the perceived pros and cons of smoking versus quitting [15,16]. As individuals progress through stages of change, the perceived benefits of quitting gradually outweigh the perceived costs, thereby facilitating behavioral change and treatment adherence [17].

While motivation is a key driver of cessation, it does not operate in isolation. Young smokers frequently present with co-occurring addictive behaviors—both substance-related and behavioral—that may alter how decisional balance is experienced. Substance use (e.g., alcohol or cannabis) and behavioral addictions (e.g., gambling, internet use, compulsive buying, or sex addiction) can intensify withdrawal symptoms, undermine treatment adherence, and increase relapse risk [18,19,20,21]. Research in this area is still limited, and few studies [22,23] have examined how comorbidities interact with motivational processes or influence preferences for app-based versus face-to-face interventions.

The present study addresses this gap by analyzing the relationship between decisional balance and co-occurring addictive behaviors in young smokers seeking treatment. Specifically, we compared participants who selected an app-based intervention with those who opted for face-to-face therapy, aiming to identify distinct motivational and behavioral profiles that can inform the design of more personalized smoking cessation programs.

## 2. Materials and Methods

### 2.1. Participants

This cross-sectional study was conducted at the University of Granada between 20 September 2021 and 22 November 2023. A total of 98 young adult tobacco users were recruited through posters, flyers, and direct outreach on campus. Participants were allocated to one of two groups according to their preferred treatment modality: 35 chose the Prescinde app-based cognitive–behavioral therapy (CBT) program, while 63 selected the traditional face-to-face CBT program.

Eligibility criteria included the following: (1) age ≥ 18 years; (2) current tobacco use, defined as daily or occasional smoking in the past month; (3) willingness to reduce or quit smoking; and (4) being enrolled as a student, staff member, or affiliate of the University of Granada. Exclusion criteria were as follows: (1) severe mental health disorders (e.g., psychotic disorders) that could compromise informed consent or participation, (2) concurrent enrollment in another smoking cessation program, and (3) lack of smartphone access (for those considering the app-based therapy).

To reduce selection bias and improve comparability between groups, a propensity score adjustment was applied [24,25]. This statistical technique estimates the probability of assignment to each treatment group based on observed baseline characteristics, thereby minimizing potential confounding in non-randomized designs. Sociodemographic variables—including age, gender, educational attainment, and academic status—were analyzed to explore their influence on treatment preference and potential effectiveness. Detailed sociodemographic data are presented in Table 1.

### 2.2. Procedure

Participants were individually screened to confirm eligibility according to the inclusion and exclusion criteria. Those meeting requirements were invited to participate and informed of their right to withdraw from the study at any point without penalty. Written informed consent was obtained prior to enrollment.

The study protocol was approved by the Research Ethics Committee of the University of Granada. After providing consent, participants freely chose between the Prescinde app-based CBT program and the face-to-face CBT program. Both treatment modalities have been described in previous research [12,13,26].

The interventions followed a standardized CBT protocol consisting of six weekly sessions (or modules) of approximately 60 minutes each. The face-to-face program included craving management, cognitive restructuring, self-control training, problem-solving, and relapse prevention, which were delivered by trained therapists following established protocols [12,13]. The Prescinde app delivered equivalent content digitally, integrating interactive exercises, personalized reminders, consumption tracking, and emotional support tools to promote adherence and self-regulation [26].

It is important to note that this study employed a cross-sectional design. The reported duration refers solely to the standard timeframe required to complete the treatment program and associated assessments; no repeated measures or long-term follow-up were conducted.

Upon completion of the intervention, all participants underwent the assessment procedures described in the following section.

### 2.3. Measures

*Nicotine dependence.* To assess physiological dependence on nicotine, we used the Fagerström Test for Nicotine Dependence (FTND) [27,28]. This six-item questionnaire explores aspects such as daily cigarette intake, how soon after waking the first cigarette is smoked, and difficulty refraining from smoking in restricted places. The total score ranges from 0 to 10, with higher scores indicating stronger nicotine dependence.

Behavioral and substance-related addictions. The MULTICAGE CAD-4 [29,30] was applied to detect a broad range of addictive behaviors. This self-report instrument includes dichotomous (yes/no) items that cover alcohol and drug consumption, gambling, eating-related problems, and other behavioral addictions. Positive responses within each domain indicate problematic involvement, making it a practical tool for screening co-occurring addictions.

*Cannabis dependence.* Cannabis use was assessed with the Severity of Dependence Scale (SDS) [31]. The SDS contains five items that evaluate difficulties in controlling consumption, worries about use, and problems with stopping. Each item is rated on a 4-point scale (0–3), producing a total score from 0 to 15. Higher scores reflect greater psychological dependence. This brief measure has shown solid psychometric properties in diverse populations [32,33,34].

*Motivation for change*. Motivation was evaluated through the Decisional Balance Questionnaire (DBQ) [35,36,37], which is based on the transtheoretical model of change. The DBQ examines perceived pros and cons of smoking and of quitting. High scores on the “pros” dimension suggest greater motivation to keep smoking, whereas high scores on the “cons” dimension reflect greater motivation to quit. Although originally designed for cannabis use, it was applied here to tobacco without item modification. This decision rests on the instrument’s theoretical foundation, which makes it suitable for studying ambivalence and readiness to change across different addictive behaviors [35,38].

### 2.4. Statistical Analyses

Descriptive statistics were computed for all study variables. Logistic regression was conducted to examine predictors of treatment choice (app-based vs. face-to-face). Stepwise multiple regression analyses were applied to explore the association between decisional balance scores and comorbid addictive behaviors. Predictors with extremely low prevalence were excluded to prevent unstable estimates, and variables exhibiting high multicollinearity were also removed to ensure model validity. Model selection was based on statistical significance and theoretical relevance. All analyses were performed using R software (version 4.4.1; R Foundation for Statistical Computing, Vienna, Austria) [39], with statistical significance set at *p* < 0.05.

## 3. Results

### 3.1. Descriptive Análisis

The final sample included 98 smokers: 63 in the face-to-face group and 35 in the app-based group. The mean age was 21.9 years (SD = 4.2). Participants who chose the app were significantly younger than those who selected face-to-face therapy (*p* < 0.01). Women made up 52% of the total sample, with no significant group differences. Most participants were undergraduate students (72.4%), and students were more likely to choose the app-based program (*p* < 0.05). Employment status did not differ between groups. Detailed sociodemographic data are presented in Table 1.

### 3.2. Predictors of Treatment Choice

Logistic regression analyses identified two significant predictors of selecting the app-based intervention: younger age (OR = 0.86, 95% CI = 0.77–0.95, *p* < 0.01) and being a student (OR = 3.25, 95% CI = 1.10–9.61, *p* < 0.05). Sex and employment status were not significant predictors. The full logistic regression model is presented in Table 2.

### 3.3. Decisional Balance and Addictive Behaviors

Stepwise regression analyses were conducted to examine associations between decisional balance (perceived disadvantages of smoking) and various addictive behaviors. Results are presented in Table 3, Table 4 and Table 5.

Table 3 shows that compulsive buying was positively associated with higher scores on the disadvantages of smoking (β = 0.26, t = 2.46, *p* = 0.016).

Table 4 displays negative associations for sex addiction (β = −0.25, t = −2.41, *p* = 0.018).

Table 5 reports additional negative predictors, including cannabis dependence (β = −0.24, t = −2.33, *p* = 0.021) and other substance-related addictions (β = −0.23, t = −2.27, *p* = 0.025).

Taken together, these models explained approximately 21% of the variance (adjusted R^2^ = 0.21), indicating that specific comorbid addictions influence motivational appraisals of smoking.

### 3.4. Complementary Analyses

Complementary regression models are reported in Table 6, Table 7 and Table 8, which provide detailed breakdowns for additional decisional balance dimensions. While compulsive buying remained the strongest positive predictor of recognizing smoking disadvantages, other addictions (particularly cannabis and sex-related problems) consistently emerged as negative predictors across models.

### 3.5. Non-Significant Findings

No significant associations were observed between nicotine dependence (FTND scores) and treatment choice, nor between alcohol-related problems and decisional balance (see Table 6, Table 7 and Table 8). These null findings suggest that treatment preferences and motivational profiles are shaped more by sociodemographic variables and specific behavioral or substance-related addictions than by nicotine dependence alone.

## 4. Discussion 

The present study examined young smokers’ preferences for smoking cessation treatment, comparing face-to-face therapy with an app-based intervention, while also considering motivational processes and comorbid addictive behaviors. Several noteworthy findings emerged.

First, younger participants and students were more likely to choose the app-based intervention. This result supports previous research indicating that digital health tools are especially attractive to younger populations, who tend to be more familiar with technology and value accessibility and flexibility [7,32]. These findings highlight the potential of mobile health applications to reduce barriers to treatment uptake in groups less inclined to attend traditional therapy.

Second, our results confirm that decisional balance is not shaped solely by tobacco use itself but also by the presence of other addictive behaviors. In particular, compulsive buying was associated with greater recognition of the disadvantages of smoking, whereas sex addiction, cannabis dependence, and other substance-related addictions were linked to a reduced perception of these disadvantages. This pattern suggests that decisional balance interacts with broader behavioral tendencies, which may either reinforce or hinder motivation to quit.

These findings are consistent with prior studies showing that compulsive or impulsive traits can influence motivation for change in complex ways [3,10,39]. Importantly, they also extend the literature by demonstrating that co-occurring addictions are not merely background factors but active variables shaping motivational profiles. For smoking cessation interventions, this means that young people struggling with multiple addictive behaviors may require additional therapeutic components to sustain motivation and adherence.

Interestingly, nicotine dependence itself was not significantly associated with treatment preference or decisional balance. This contrasts with earlier studies where dependence severity predicted treatment outcomes [13,22]. A plausible explanation is the relatively young age of our sample, with shorter smoking histories and lower overall dependence. In such cases, motivational and contextual variables appear to play a more decisive role than nicotine dependence alone.

From a clinical perspective, these findings underscore the need to tailor smoking cessation programs to the individual. Digital interventions seem particularly suitable for younger student populations, while those with co-occurring addictions may benefit from more comprehensive approaches that also target impulsivity, compulsivity, and emotional regulation. Future work should therefore consider integrated models that combine smoking cessation with interventions for co-addictive patterns, ensuring that treatment addresses the broader behavioral context of each individual.

## 5. Strengths and Limitations 

The main strengths of this study include its focus on a vulnerable and understudied population—young smokers with comorbid addictions—and its use of both logistic and regression models to capture treatment predictors and motivational processes. However, several limitations should be noted. First, the sample size was modest, which may have limited statistical power for detecting weaker effects. Second, the cross-sectional design precludes causal inference regarding the relationships between comorbid addictions and decisional balance. Third, self-report measures may be subject to recall or social desirability bias. Finally, this study was conducted in a specific cultural context, which may limit generalizability to other populations.

## 6. Conclusions 

This study highlights the importance of tailoring smoking cessation strategies to the psychological and behavioral profiles of young smokers. Co-occurring addictive behaviors significantly shape both the perception of tobacco-related disadvantages and the choice of intervention. Digital therapies, such as the Prescinde app, appear especially attractive and accessible for younger individuals, while face-to-face interventions may be more appropriate for those with complex clinical profiles or multiple comorbidities.

These findings support the value of adopting personalized and flexible approaches to treatment, ensuring that programs align with users’ motivational profiles, behavioral patterns, and specific support needs. Future research should explore the long-term effectiveness of each modality and assess hybrid models that integrate digital tools with in-person support. Such approaches may optimize engagement, adherence, and cessation outcomes, particularly in young populations facing multiple addictive behaviors.

## 7. Implications and Future Directions 

The results suggest that smoking cessation programs targeting young adults should prioritize digital delivery formats while systematically screening for co-occurring addictive behaviors that can undermine motivation. Longitudinal studies are needed to evaluate whether app-based interventions maintain their efficacy over time and to clarify the causal links between comorbid addictions, decisional balance, and cessation outcomes. Expanding research to more diverse populations will further strengthen the generalizability and clinical utility of these findings.

## Figures and Tables

**Table 1 healthcare-13-02326-t001:** Sociodemographic characteristics and prevalence of co-occurring addictive behaviors by intervention type.

Variable	App-Based Therapy (n = 35) Mean/%	Face-to-Face Therapy (n = 63) Mean/%	Contrast Statistic	*p*-Value
Gender: Female (%)	65.7%	44.4%	3.271	n.s.
Gender: Male (%)	34.3%	55.6%		
Education: Degree/Bachelor’s (%)	25.7%	36.5%	2.295	n.s.
Education: Master’s/PhD (%)	17.1%	22.2%		
Education: Primary/Secondary (%)	57.1%	41.3%		
Currently Studying: Yes (%)	71.4%	42.9%	6.272	<0.05
Currently Studying: No (%)	28.6%	57.1%		
Age (Years)—Mean ± SD	26.69 ± 11.75	40.43 ± 14.84	−4.715	<0.001
Alcohol Abuse (%)	6.5%	2.6%		<0.05
Drug Use (%)	3.7%	1.5%		<0.001
Pathological Gambling (%)	0.4%	0.0%		n.s.
Eating Disorders (%)	3.7%	0.8%		n.s.
Internet Addiction (%)	5.2%	0.3%		<0.01
Video Game Addiction (%)	1.0%	0.2%		<0.05
Compulsive Buying (%)	1.6%	0.2%		<0.05
Sex Addiction (%)	1.5%	0.5%		n.s.
Cannabis Dependence (%)	8.4%	9.6%		n.s.

Note: Percentages are based on the proportion of participants in each group. Contrast statistic is from chi-square test (categorical variables) or t-test (continuous variables). *p*-values for addictive behaviors are from Mann–Whitney–Wilcoxon tests. n.s. = not significant.

**Table 2 healthcare-13-02326-t002:** Logistic regression predicting type of intervention (app-based = 1, face-to-face = 0).

Variable	Estimate	Odds Ratio	Std. Error	z-Value	*p*-Value
(Intercept)	−1.3300	0.264	0.8286	−1.6050	n.s.
Age	0.0922	1.097	0.0255	3.6190	<0.001
Alcohol Abuse	−0.5361	0.585	0.2408	−2.2260	<0.05
Pathological Gambling	−18.2470	0.000	1536.5840	−0.0120	n.s.
Compulsive Buying	−1.2626	0.283	0.5656	−2.2320	<0.05
Cannabis Dependence	0.1845	1.203	0.0904	2.0410	<0.05

n.s. = non-significant.

**Table 3 healthcare-13-02326-t003:** Stepwise regression for decisional balance (advantages)—app-based therapy.

Variable	Estimate	Std. Error	z-Value	*p*-Value
(Intercept)	16.5875	2.3455	7.0720	<0.001
Pathological Gambling	−3.1198	1.2151	−2.5670	<0.05
Substance Addiction	−1.3372	0.8397	−1.5920	n.s.
Eating Disorders	1.5335	0.9546	1.6070	n.s.
Internet Addiction	1.4243	1.0312	1.3810	n.s.
Cannabis Dependence	0.6509	0.3377	1.9270	n.s.

n.s. = not significant.

**Table 4 healthcare-13-02326-t004:** Stepwise regression for decisional balance (advantages)—face-to-face therapy.

Variable	Estimate	Std. Error	z-Value	*p*-Value
(Intercept)	6.0841	2.5416	2.394	<0.05
Nicotine Dependence	2.0904	0.5176	4.038	<0.001
Substance Addition	1.8714	0.9138	2.048	<0.05
Eating Disorders	3.6416	1.2443	2.927	<0.001
Compulsive Buying	8.6446	4.0181	2.151	<0.05
Sex addiction	−6.1635	1.8848	−3.27	<0.001

**Table 5 healthcare-13-02326-t005:** Stepwise regression for decisional balance (advantages)—total sample.

Variable	Estimate	Std. Error	z-Value	*p*-Value
(Intercept)	13.8768	2.2021	6.301	<0.001
Nicotine Dependence	0.7241	0.3875	1.869	n.s.
Pathological Gambling	−3.5834	1.2292	−2.915	<0.01
Eating Disorders	2.1661	0.6943	3.12	<0.01
Internet Addiction	0.8146	0.5807	1.403	n.s.
Cannabis Dependence	0.4119	0.2087	1.974	n.s.
Type of intervention	−2.5783	1.7053	−1.512	n.s.

n.s. = not significant.

**Table 6 healthcare-13-02326-t006:** Stepwise regression for decisional balance (disadvantages)—app-based therapy.

Variable	Estimate	Std. Error	z-Value	*p*-Value
(Intercept)	23.9720	1.4720	16.2870	<0.001
Compulsive Buying	3.6990	1.2570	2.9430	<0.01

**Table 7 healthcare-13-02326-t007:** Stepwise regression for decisional balance (disadvantages)—face-to-face therapy.

Variable	Estimate	Std. Error	z-Value	*p*-Value
(Intercept)	27.3531	1.0717	25.523	<0.001
Alcohol Abuse	1.3192	0.4708	2.802	<0.01
Substance Addiction	−2.6521	0.6225	−4.261	<0.001
Eating Disorders	3.999	0.9691	4.127	<0.001
Internet Addiction	−1.0897	0.5098	−2.137	<0.05
Compulsive Buying	6.7116	3.2582	2.06	<0.05
Sex Addiction	−5.6482	1.4913	−3.787	<0.001

**Table 8 healthcare-13-02326-t008:** Stepwise regression for decisional balance (disadvantages)—total sample.

Variable	Estimate	Std. Error	z-Value	*p*-Value
(Intercept)	26.712	1.1632	22.965	<0.001
Alcohol addiction	0.8379	0.48	1.746	n.s.
Eating Disorders	1.0118	0.6466	1.565	n.s.
Video Game Addiction	−1.7558	0.9371	−1.874	n.s.
Compulsive Buying	2.054	0.9922	2.07	<0.05
Sex Addiction	−3.5985	0.887	−4.057	<0.001
Cannabis Dependence	−0.4201	0.1889	−2.224	<0.05

n.s. = not significant.

## Data Availability

The datasets presented in this article are not readily available because the data are part of an ongoing study.

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
