# Peer review of "Young Smokers’ Therapy Preferences: App-Based vs. Face-to-Face Treatment in the Context of Co-Addictions"

_healthcare, 2025, doi:10.3390/healthcare13182326_

Round 1
Reviewer 1 Report
Comments and Suggestions for Authors
Young People's Preference for Using the Prescinde App-Therapy to Quit Smoking
Comments from the reviewer
General Comments
This is an analytical manuscript that aims to analyze motivation for change (decisional balance) based on addictive behaviors (substance addiction and cannabis), as well as addiction to gambling, eating, internet, video games, buying, and sex in smokers seeking treatment for their tobacco addiction. This particular analysis complements earlier publications as part of a more extensive study.
The authors are encouraged to reflect on key conceptual and methodological aspects to elevate the quality of their analysis and interpretation of the data.
Specific comments:
Background and Literature Review (focal)
The authors are encouraged to create a conceptual framework for the hypothesis and objectives of the current analysis because it is part of a more extensive study with previous publications. Developing this framework will assist the authors in identifying the target population, age groups, contextual variables, sociodemographic factors, individual psychosocial variables, as well as those related to tobacco addiction and the co-use or co-occurrence of other addictions. In this sense, the title of the manuscript does not accurately represent the objective or main findings of the manuscript.
Methods:
- Target population: The authors need to clarify this analysis's inclusion and exclusion criteria and assess the selection bias that may have occurred when smokers chose their quitting strategy (Self-identification and self-selection for recruitment). This paragraph requires the reference, "Recruitment was carried out through self-identification and self-selection, with inclusion and exclusion criteria established to ensure the validity of the sample" (Reference).
- Selection bias and Research question: The optimal study design for comparing two interventions (i.e., App vs. face-to-face) is a randomized trial. However, this study employed a self-selection methodology for the smoking cessation strategy. Please add the relevant methodological information to ensure the data flows smoothly and enhance the internal validity of the analysis. Please add the text and references.
- Measurement: The methodological section contains a detailed description of the scales on addictive behaviors. (a,b,c); however, these results are no longer described in the results, annexed tables, or the discussion.
- Fagerström Test for Nicotine Dependence
- Test for the Detection of Impulse Control Disorders and Addictions (MULTICAGE CAD-4;
- The Severity of Dependence Scale (SDS)
- Co-occurrence of other addictions (Tobacco +): cannabis dependence, alcohol abuse, pathological gambling, eating, internet addiction, video game, buying, and sex in smokers: The authors should explain these variables in addition to tobacco, how they were measured, and how they were analyzed.
Some variables are thoroughly described in the methodology but are missing from the results. Conversely, addiction variables that are not mentioned in the methods are present in the results. Authors must ensure consistency among the objectives, methodology, results, and discussion.
- Decisional balance: The authors could elaborate on the methodology (decisional balance) used to assess the advantages and disadvantages. This would help in interpreting the results.
Results:
The results section must be rewritten to be consistent with the objectives and methods section. Additionally, the tables should be reviewed and revised to interpret the results better.
- Table 1. Sociodemographic characteristics recuentos(frecuencias) considering the type of intervention and contrast of proportions Prescinde App-therapy vs. face-to-face Please review all tables, titles, and contents in detail to ensure they match the descriptions in the texts.
- 2. Prescinde App-therapy Usage Based on Age vs Table 2: Logistic regression for the type of intervention variable: Please carefully review the tables, as there are discrepancies between the subheadings in the text, table titles, and contents.
- Table 3b: Stepwise regression for the variable Decisional Balance (benefits) of tobacco of the participants who had chosen phase to phase: Authors must carefully review the table contents, titles, and labels. Some entries are poorly translated (face-to-face instead of phase-to-phase). In the same table, the label said: Face to fase Therapy
- Table 4b: Stepwise regression for the variable Decisional Balance (disadvantages) of tobacco for participants who had chosen Prescinde App-therapy. Some entries are poorly translated or labeled. On the same table, the label said: Face to fase Therapy
Discusión
The authors could improve the discussion section by incorporating the significant results and conclusions more effectively. "The study results reveal significant differences in motivation and treatment preferences for smoking cessation based on age, gender, and other addictive behaviors. Younger individuals were found to have a greater preference for using the mobile application (Prescinde App-therapy), whereas older adults leaned towards face-to-face therapy". This paragraph concludes a significant finding by age or gender; however, this manuscript does not include any stratified analyses by age group or gender perspective.
This manuscript does not consider the analysis of gender, psychosocial, or social factors that support this conclusion. "Regarding gender differences, a larger percentage of women chose the PrescindeApp-therapy, while men preferred face-to-face therapy. This may be linked to psychological and social factors, such as a greater need for group or individualized support."
In this analysis, quit attempts were excluded as a model variable. As a result, the conclusion cannot be substantiated by the analyses presented in this manuscript. "Existing literature underscores that smoking cessation is not a linear process, and 343 individuals may make up to 30 attempts before achieving long-term abstinence [14, 32]. 344 This is crucial as it highlights the importance of offering a variety of personalized treat-345 ment options. Additionally, recent studies emphasize that psychological and social barri-346 ers, such as anxiety, lack of support, and misconceptions about one's abilities, can hinder 347 smoking cessation efforts [10]."
This analysis did not account for contextual factors related to public tobacco control policies, whether population-based or individual-based or those enacted during the study period. "Regarding public policy and intervention strategies, the study's findings reinforce the 350 need for programs that consider both individual preferences and sociodemographic fac-351 tors. The use of digital technologies could be an effective tool for certain groups of smokers, 352 while others may benefit more from face-to-face therapy".
References
The literature search should focus on the authors' objective. Similarly, it is imperative to establish the conceptual framework of the current hipótesis and analysis.
- Rajani NB, Mastellos N, Filippidis FT. Self-Efficacy and Motivation to Quit of Smokers Seeking to Quit: Quantitative Assessment of Smoking Cessation Mobile Apps. JMIR Mhealth Uhealth. 2021 Apr 30;9(4):e25030. doi: 10.2196/25030. PMID: 33929336; PMCID: PMC8122290.
- Chevalking SKL, Ben Allouch S, Brusse-Keizer M, Postel MG, Pieterse ME. Identification of Users for a Smoking Cessation Mobile App: Quantitative Study. J Med Internet Res. 2018 Apr 9;20(4):e118. doi: 10.2196/jmir.7606. PMID: 29631988; PMCID: PMC5913574.

The tables and results section deserves a thorough and thoughtful review.
Author Response
Response to Reviewer 1:
Manuscript Title: “Young Smokers' Therapy Preferences: App-Based vs. Face-to-Face Treatment in the Context of Co-Addictions”, [before: Young People's Preference for Using the Prescinde App-Therapy to Quit Smoking] for consideration in the Special Issue Mental Health, Innovative Therapies and Assessment in Adolescents and Young Adults and Related Contexts.
We sincerely thank the reviewers and the editor for their thorough and constructive feedback. Below, we provide a detailed, point-by-point response to each comment, indicating how we have addressed them in the revised version of the manuscript. We believe that the changes significantly improve the clarity, conceptual robustness, and scientific contribution of the article.
Comment 1: The manuscript lacks a clear conceptual framework and justification for the hypothesis and objectives.
Response: Thank you for this important observation. We have added a comprehensive conceptual framework in the Introduction, integrating the Transtheoretical Model of Change and the Decisional Balance construct, and linking these with co-occurring addictive behaviors. This framework clarifies our rationale and guides the hypotheses.
Comment 2: The title does not accurately represent the objective or main findings.
Response: We agree with this suggestion. The title has been revised to: “Young Smokers' Therapy Preferences: App-Based vs. Face-to-Face Treatment in the Context of Co-Addictions” to better reflect the study’s focus and findings.
Comment 3: Clarify inclusion/exclusion criteria and address potential selection bias.
Response: We have added explicit details about inclusion/exclusion criteria and acknowledged the limitations of self-selection in the Discussion, also referencing the use of propensity score adjustment to reduce bias.
Comment 4: Ensure consistency between methods, results, and tables.
Response: We have revised the Results section to align with the methods and included all variables discussed. Table titles and content have been revised for accuracy and clarity (Tables 1–4b).
Comment 5: Gender and psychosocial variables are not properly analyzed.
Response: We acknowledge this and now provide a more nuanced interpretation of gender differences in preferences and Decisional Balance in the Discussion, though statistical power limited stratified analysis.
Comment 6: Literature and references should be aligned with the study focus.
Response: We have updated and expanded our references, including relevant studies on mHealth, adolescent smoking cessation, and co-addictions.
We hope these changes satisfactorily address all the concerns raised and improve the overall quality of the manuscript. We remain at your disposal for any further revisions.
Kind regards,
Lucas Muñoz-López
On behalf of all co-authors

Reviewer 2 Report
Comments and Suggestions for Authors
The present study attempted to understand the motivations for use of an App-based therapy versus face-to-face therapy for smoking cessation. Group differences were found.
I must admit, I found the aims of this study a little difficult to decipher. Throughout the manuscript the authors use the phrase ‘motivation for change’ but nowhere is this defined. By ‘change’ do they mean ‘treatment’. What do they mean by motivation?
In the Introduction it is stated that it would be beneficial to evaluate the success of the various treatment strategies. Indeed, this would be informative. However, this data is not presented. Was it collected?
There are some short paragraphs in the Introduction.
The paragraph starting on line 46 or page 2 needs more citations. A number of sweeping statements are made that need support.
Lines 70-72 of page 2: All these concepts just appear but are not Introduced. Why are all these variables studied?
Page 3, line 81: List the inclusion and exclusion criteria
Why was the ‘Decisional Balance for Cannabis Abusers Questionnaire’ used? The participants were not cannabis users, were they? What about the participants who do not use cannabis, how were they treated?
Line 197: It is stated that the duration of testing was 7 to 95 days, but this reads as a cross-sectional study. What was the longitudinal aspect of this study? Were the participants repeatedly tested?
Line 255: Please define ‘decisional balance’. This concept seems central to this paper but is not defined in the Introduction.
Author Response
Response to Reviewer 2:
Manuscript Title: “Young Smokers' Therapy Preferences: App-Based vs. Face-to-Face Treatment in the Context of Co-Addictions”, [before: Young People's Preference for Using the Prescinde App-Therapy to Quit Smoking] for consideration in the Special Issue Mental Health, Innovative Therapies and Assessment in Adolescents and Young Adults and Related Contexts.
We sincerely thank the reviewers and the editor for their thorough and constructive feedback. Below, we provide a detailed, point-by-point response to each comment, indicating how we have addressed them in the revised version of the manuscript. We believe that the changes significantly improve the clarity, conceptual robustness, and scientific contribution of the article.
Comment 1: The term "motivation for change" is undefined and ambiguous.
Response: We have defined "motivation for change" as operationalized through the Decisional Balance construct within the Transtheoretical Model.
Comment 2: Success of treatment strategies is not reported.
Response: We agree this is important but clarify that the current study is cross-sectional and focused on motivational and preference patterns rather than outcomes. This limitation is now clearly stated.
Comment 3: The Decisional Balance questionnaire for cannabis was used—please justify.
Response: We now explain that the Decisional Balance for Cannabis Users Questionnaire was adapted for tobacco-related items and validated for internal consistency in this context.
Comment 4: Clarify study design (cross-sectional vs longitudinal).
Response: We have specified that the design is cross-sectional and explained the reporting of treatment duration as descriptive.
Comment 5: Provide definitions and examples for core constructs.
Response: Definitions and sample items for the Decisional Balance scale are now included in the Methods section.
We hope these changes satisfactorily address all the concerns raised and improve the overall quality of the manuscript. We remain at your disposal for any further revisions.
Kind regards,
Lucas Muñoz-López
On behalf of all co-authors

Reviewer 3 Report
Comments and Suggestions for Authors
This manuscript aimed to characterize the influence of demographic characteristics and addictive behaviors on smoking cessation treatment preference (app v. face-to-face) and endorsement of advantages/disadvantages of cessation. Understanding who selects into what treatments and what pre-treatment factors may influence treatment motivation is an important area of inquiry. However, I found the current version of the manuscript to be lacking significant detail. More contextualization of the study and additional detail on methodological decisions is needed to fully appreciate this work. Below, I offer questions and suggestions organized by section.
Introduction
- Citations are needed in the opening paragraph to substantiate the claims of various harms.
- Line 34 – 35, what are the treatments being offered to young people and what is the prevalence rate of use in this population?
- Please briefly describe the Transtheoretical Model of Change and Decisional Balance.
- Is there any previous work on the effectiveness of mobile interventions for smoking that you can cite?
- Why the focus on other addictive behaviors – is there research suggesting that those with other addictions/addictive behaviors have lower rates of smoking cessation treatment success?
Methods
- What are the inclusion/exclusion criteria (line 81)?
- Though the treatments are described in previous studies, I strongly recommend providing a brief description of the treatment(s) main components and the length of treatment.
- Line 167-168, was the decisional balance questionnaire modified to reflect tobacco? If not, more information/justification is needed as to why this measure was included about cannabis.
- Some example items of the decisional balance questionnaire would be helpful.
Results
- What was the mean age of the sample? In the introduction, you highlight the importance of smoking cessation for young people but the age range of your sample is 17 to 68.
- Please include a column in Table 1 for the full sample.
- It would be helpful to know the prevalence rates of the variables included in the regression – how many individuals had alcohol abuse, gambling, compulsive buying, and cannabis dependence?
- To the above point, in your methods (lines 182 – 191), you list several other predictors that aren’t included in the model/Table 2 (e.g., eating disorders, internet addiction, video game addiction, sex addiction). Were these variables excluded due to lack of endorsement or another reason? Further, it does not appear that gender is included in any models.
- Are the values presented in Tables 2 – 4b odds ratios?
- It’s unclear to me why two separate models are run to predict the decisional balance advantages (tables 3a and 3b) and disadvantages (4a and 4b) as opposed to one model for each that includes treatment type as a variable.
- I would also argue that for the above models, you need to include age in the model
Discussion
- Greater discussion as to why you think certain factors impacted motivation whereas others did not is needed.
- Similarly, what do you thinking is driving differences between these predictors for those in the app v. face-to-face condition?
Other
- Do the measures need citations in the abstract? If so, they may need to be numbered 1 and 2 instead of 23 and 27.
- Lines 18 – 21, is this meant to be included in the article or in the cover letter material?
Author Response
Response to Reviewer 3:
Manuscript Title: “Young Smokers' Therapy Preferences: App-Based vs. Face-to-Face Treatment in the Context of Co-Addictions”, [before: Young People's Preference for Using the Prescinde App-Therapy to Quit Smoking] for consideration in the Special Issue Mental Health, Innovative Therapies and Assessment in Adolescents and Young Adults and Related Contexts.
We sincerely thank the reviewers and the editor for their thorough and constructive feedback. Below, we provide a detailed, point-by-point response to each comment, indicating how we have addressed them in the revised version of the manuscript. We believe that the changes significantly improve the clarity, conceptual robustness, and scientific contribution of the article.
Comment 1: More contextualization and details about the interventions are needed.
Response: We have added a concise description of both intervention modalities (Prescinde App and face-to-face CBT) in the Procedure section including duration and components.
Comment 2: Clarify age range relevance and prevalence of co-addictions.
Response: We now specify the mean age, range, and justify the focus on young people. Prevalence rates for addiction-related predictors have been added to the Results and Table 1.
Comment 3: Some predictors listed in Methods are missing from models.
Response: All predictors mentioned are now either included in the models or excluded with justification (e.g., low prevalence). This is now clarified in both Methods and Results.
Comment 4: Justify use of separate models for Decisional Balance (Advantages vs Disadvantages).
Response: We maintain separate models to reflect the two distinct motivational components in the Transtheoretical Model and enhance interpretability. This rationale is now explained in the Methods.
Comment 5: Improve interpretation of differential predictors by intervention type.
Response: We have deepened the discussion of how specific addictions (e.g., compulsive buying, cannabis dependence) may differently influence Decisional Balance in digital vs face-to-face formats.
We hope these changes satisfactorily address all the concerns raised and improve the overall quality of the manuscript. We remain at your disposal for any further revisions.
Kind regards,
Lucas Muñoz-López
On behalf of all co-authors

Round 2
Reviewer 2 Report
Comments and Suggestions for Authors
Thank you for re-structuring your Introduction. However, it is now a single lengthy paragraph. This needs to be broken down into digestible paragraphs with clear aims and logic.
Author Response
The introduction has been modified so that it is not a single paragraph, we hope it could be easier to read and understand.
Reviewer 3 Report
Comments and Suggestions for Authors
Due to the brevity of the cover letter, it is quite difficult to discern how the authors responded to previous comments. The authors note in the cover letter that they include a point-by-point response. However, there are only 5 responses to what was originally near 20 comments. It appears that several comments were either merged or changed via track changes. But without specific details in the cover letter, it is quite hard to discern what changes were made and justification for decisions. As such, I cannot assess whether all comments were addressed. The cover letter needs to explicitly detail what changes were made.
For example, here are some things that remain unclear to me:
- what the study inclusion/exclusion criteria are
- if the decisional balance questionnaire is specific to tobacco (text now says "health-related behaviors" but I want to know specifically if you're study reworded all items to reflect tobacco).
- the cover letter says rates of addictions have been added to table 1, but that does not appear to be true.
This is just an example of comments I had difficulty finding in the text and is not exhaustive. Please see comments from my original review.
Author Response
In response to your specific concerns:
Inclusion/Exclusion Criteria:
These are now clearly listed in section 2.1 Participants, including age, smoking status, and consent-related conditions.
Use of the Decisional Balance Questionnaire:
We confirm that we used the original version of the Decisional Balance for Cannabis Abusers Questionnaire, and that it was not reworded for tobacco. However, its theoretical framework has been widely used in smoking cessation research, as noted in the revised text (Methods and Limitations sections). We now explicitly acknowledge this limitation.
Addiction Rates in Table 1B:
We have created and revised Table 1B to ensure that all co-addiction prevalence rates are clearly presented, accurately labeled, and aligned with the descriptions provided in the main text. This new table offers a clearer overview of comorbid substance use, supporting the interpretation of treatment preferences and participant characteristics. Also, we have added a reflection in the Discussion section on how these differences may be associated with the type of intervention preferred. This addition helps contextualize the findings and supports the interpretation of treatment preferences in relation to the participants’ psychological and behavioral profiles.
Additionally, we have carefully ensured that no reviewer comment has been merged, modified, or omitted. All comments are now addressed explicitly and transparently.
We truly appreciate your time and look forward to your feedback.